# When Fine-Tuning Fails and When It Generalises: Data Diversity, Mix Training, and GGUF Efficiency in LLM-based TTS

## Abstract

Large language models are increasingly adopted as semantic backbones for neural text-to-speech systems. However, frozen LLM representations are insufficient for modeling speaker-specific acoustic and perceptual characteristics. Our experiments involving fine tuning of the Language Model backbone of TTS show promise in improving the voice consistency and Signal to Noise ratio (SNR) in voice cloning task. Across multiple speakers, LoRA fine-tuning consistently outperforms the non–fine-tuned base Qwen-0.5B model across three complementary dimensions of speech quality. First, perceptual quality improves significantly, with DNS-MOS gains of up to +0.42 points for speakers whose training data exhibits sufficient acoustic variability. Second, speaker fidelity improves for all evaluated speakers, with consistent increases in voice similarity, indicating that LoRA effectively adapts speaker identity representations without degrading linguistic modeling. Third, signal-level quality improves in most cases, with signal-to-noise ratio increasing by as much as 34 percent. Crucially, these improvements are strongly governed by the characteristics of the training data. Speakers with high variability in acoustic energy and perceptual quality achieve simultaneous gains in DNS-MOS, voice similarity, and SNR. In contrast, speakers trained on acoustically homogeneous data experience limited gains or perceptual degradation, even when voice similarity improves. This reveals that LoRA can faithfully clone speaker identity while also amplifying noise characteristics and recording artifacts present in narrow training distributions. We further identify a loss–quality divergence phenomenon in which training and validation loss continue to improve during fine-tuning while perceptual quality degrades for low-variability speakers. Besides, we show that optimal inference temperature of the language model backbone depends on training data variability, with conservative sampling benefiting low-variability speakers but degrading quality for high-variability ones.

Overall, this work establishes that LoRA fine-tuning is not merely a parameter-efficient optimization technique but an effective mechanism for better speaker-level adaptation in compact LLM-based TTS systems. When supported by sufficiently diverse training data, LoRA-adapted Qwen-0.5B consistently surpasses its frozen base model in perceptual quality, speaker similarity with low latency using GGUF model hosted in quantized form.

## 1 Introduction

Voice to voice architecture are integral to multi-modal agent development. Different voice to voice architectures viz. cascaded, end to end and hybrid architectures have been proposed. Cascaded systems are easier to build initially whereas end-to-end requires more specialized expertise. End-to-end native voice models are being prioritized for consumer applications where naturalness and latency matter most, while cascaded approaches remain popular in enterprise settings where tool calling and advanced reasoning capabilities provided by frontier LLMs are priority. Cascading multi-modal agents which support voice input and output, comprise of three principle components, Automated Speech Recognition (ASR), Text based Agent (LLM) and Text-to-Speech (TTS), each of which are susceptible to errors of different kinds. TTS powered by language model backbones have shown promise in using Language model backbone to predict sequences of acoustic tokens conditioned on text and a speaker prompt Choudhary and Purwar (2025) Neuphonic (2025)

TTS (2025) Wang et al. (2023) GPT-SoVITS (2025). Language model based TTS systems collapse linguistic modeling, prosody planning, and long-range acoustic coherence into a single autoregressive backbone. However, despite their LM-centric design, existing literature does not report systematic LoRA fine-tuning of these language model backbones for TTS adaptation, leaving its impact on perceptual audio quality in voice cloning largely unexplored.

Parameter-efficient fine-tuning (PEFT) methods, particularly Low-Rank Adaptation (LoRA), have emerged as a practical solution for adapting large text-to-speech (TTS) models under memory and latency constraints. Existing LoRA-based TTS literature predominantly applies adaptation to downstream synthesis components viz. acoustic decoders, speaker embeddings, or style and emotion control modules, while keeping the linguistic or semantic modeling backbone frozen. Representative works including LoRP-TTS, StyleSpeech, EELE, and LoRA-based multi-speaker VITS adaptations follow this paradigm, focusing on efficient control or personalization without modifying the core generative model (Table 1). UtterTune represents one of the only reported research which inject LoRA into LM layers; however, its scope is primarily limited to pronunciation and pitch-accent control, without broader analysis of perceptual quality, inference latency, or stability trade-offs (Table 1).

As a result, several fundamental questions remain unanswered in the literature. First, it is unclear how LoRA adaptation of an LM-based TTS backbone interacts with the pretrained acoustic prior encoded in large-scale speech-trained language models. Second, the reliability of validation loss as a proxy for perceptual quality parameters like MOS, SNR, and speaker similarity has not been rigorously examined in token-level generative TTS models. Third, the role of training data characteristics, such as acoustic variability, energy spread, and linguistic diversity, in determining the success or failure of LM-level LoRA adaptation is poorly understood. Finally, while inference-time decoding controls such as temperature and top-$k$ sampling are known to influence perceptual outcomes in language models, their joint optimization with LoRA-adapted LM backbones for TTS has not been systematically studied, particularly in the context of balancing naturalness, stability, and latency.

In contrast to prior approaches summarized in Table 1, our work directly addresses these gaps. First, by conducting a comprehensive empirical study of LoRA fine-tuning applied to the Qwen-0.5B language model backbone for TTS. Next, we evaluated its performance across multiple datasets and speakers, with following main contributions:

- **LM-Backbone LoRA for TTS:** We apply LoRA directly to attention layers of a language model backbone (Qwen-0.5B) used for acoustic token prediction, moving beyond synthesis-layer-only adaptation.

- **Loss–Quality Decoupling Analysis:** We identify and characterize a failure mode where validation loss improves monotonically while perceptual quality (DNS-MOS) degrades, challenging conventional early-stopping criteria for LM-based TTS.

- **Training Data Variability Study:** Through controlled experiments across speakers and datasets, we demonstrate that acoustic variability, is a strong predictor of successful LM-level LoRA adaptation.

- **Hyperparameter Optimization:** We show that decoding hyperparameters (temperature and top-$k$ sampling) can partially mitigate perceptual degradation induced by LoRA adaptation, enabling post-training control over quality stability trade-offs.

- **Latency Optimization:** We show that model quantization and use of GGUF to store model weights, enables faster inference by significantly reducing first chunk latency.

## 2 Methodology

We performed fine tuning of qwen 0.5 billion which forms the language model backbone of NeuTTS. After fine tuning, we performed an exhaustive analysis of the effect of fine tuning on voice quality.

### 2.1 Full Finetuning

The dataset comprised of audio files along with the transcripts, the data was exclusive to a single speaker in order to observe the learning of the model to a particular speaking style. Full Finetuning was conducted on the entire set of parameters, leading to higher memory utilization. Considering training on low resource GPUs, we were constrained to

Table 1: Comparison of LoRA-based Methods in Text-to-Speech Systems

| Paper | What was implemented (2-line summary) | LoRA on LLM TTS Backbone |
|---|---|---|
| **LoRP-TTS** Łukasz Bondaruk and Kubiak (2025) Bondaruk & Kubiak (2025) | Applies LoRA to TTS model layers for rapid speaker personalization from a single noisy reference utterance. Low-rank updates adapt speaker characteristics while keeping the synthesis backbone frozen. | No |
| **TTS-Hub** Anonymous (2025) OpenReview (ICLR submission) | Introduces modular LoRA adapters trained for individual speech attributes and composed arithmetically. Enables controllable TTS without modifying the underlying language or synthesis backbone. | No |
| **StyleSpeech** Lou et al. (2024) Lou et al. (ACM MM Asia 2024) | Employs parameter-efficient LoRA fine-tuning on pre-trained TTS components for phoneme and style control. Focuses on efficiency and controllability rather than LM adaptation. | No |
| **LoRA-based Multi-Speaker TTS** IEEE Access (2024) | Integrates LoRA and residual adapters into VITS-based multi-speaker TTS architectures. Achieves efficient speaker adaptation with substantially fewer trainable parameters. | No |
| **EELE** Qi et al. (2024) arXiv:2408.10852 | Applies LoRA modules to emotional and expressive control layers in TTS systems. Targets parameter-efficient emotion adaptation without altering linguistic modeling layers. | No |
| **UtterTune** Kato (2025) (2025) | Injects LoRA directly into the Transformer-based language model backbone of an LLM-TTS system. Enables pronunciation and pitch-accent control by adapting semantic/phoneme prediction layers. | Yes |

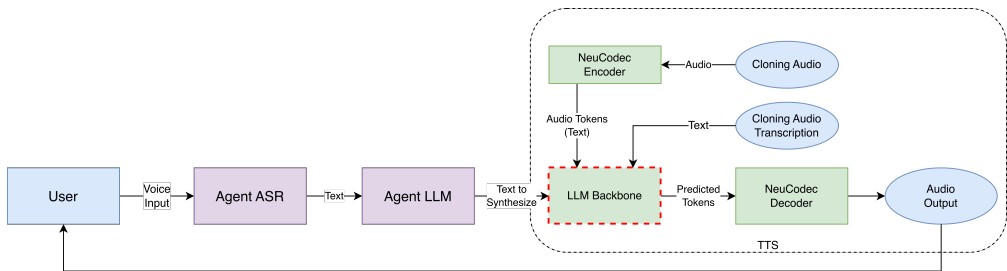

Figure 1: End-to-End Pipeline of a cascading voice agent comprising of a ASR module to transcribe voice to text, LLM module to reason over the text and generate text output and lastly a TTS module to synthesize speech for the reasoned text. Our work to the pipeline focuses on finetuning the LLM backbone (enclosed in red) responsible for token prediction, which are subsequently decoded by the neural codec to generate the waveform

keep a batch size of 2 which may have introduced instabilities in the training process. Training was performed for 5 epochs on NVIDIA L4 GPU (24GB VRAM) using Stochastic gradient descent with Adam optimizer.

## 2.2 LoRA Finetuning

We also performed LoRA Hu et al. (2021) Fine-tuning considering the memory requirements and the need for fast adaptation to a particular speaking style. LoRA with rank 8 and alpha 16 reduced the number of trainable parameters allowing use of batch size 4 and gradient accumulation of 2 leading to an effective batch size of 8. LoRA finetuning was performed on attention layers, in particular q_proj, k_proj, v_proj. Training with LoRA had a much more stable decrease of training loss over the training steps. Similar to full finetuning, the dataset was structured to contain audio files and transcripts and the training process was carried for 5 epochs.

## 2.3 Datasets

We have used audio data curated from following two sources to perform fine tuning of qwen 0.5b model.

- HiFi TTS: HiFi-TTS Bakhturina et al. (2021) includes audio data and transcripts based on public audiobooks from LibriVox and texts from Project Gutenberg. This is the source of data for speaker IDs 1, 2 and 11614.

- Libriheavy-HQ: Libriheavy-HQ Thornbury, Bryan and Mythic Infinity Labs (2024) Kang et al. (2023) is the source for data corresponding to speaker IDs 1401, 1212, 1259. On an average, the average length of audio clips in the data is lesser than the data in HiFi TTS dataset. Hence with a comparable count of files, the total length of data in the dataset is much higher.

Acknowledging the need for datasets segregated by speaker IDs for fine-tuning on a particular speaker's voice, we plan to release open-source datasets on HuggingFace conforming to this practice to support further experiments on fine-tuning of TTS models to a particular voice.

Table 2: Summary of the datasets used

|  | Data Len (hr.) | Avg Len (s) | Std. Dev. (s) | Unique Words | Avg Words |
|---|---|---|---|---|---|
| 1 | 3.523 | 2.536 | 1.239 | 4492 | 6.717 |
| 2 | 3.907 | 2.813 | 1.633 | 7981 | 7.521 |
| 11614 | 3.860 | 2.778 | 1.471 | 7460 | 8.069 |
| 1401 | 17.590 | 14.314 | 6.209 | 16064 | 41.399 |
| 1212 | 18.492 | 15.543 | 6.446 | 13067 | 42.575 |
| 1259 | 13.652 | 14.675 | 6.202 | 11943 | 42.978 |

## 2.4 Metrics

- Voice Similarity: For Text-to-Speech (TTS) systems, cloning a reference audio sample has become an important use-case for serving virtual assistants. Hence, speaker similarity in terms of audio generated being similar to the reference audio emerges as an important metric. We utilize wespeaker's embeddings and use cosine-similarity followed by a linear transformation to 0-1 scale as the metric for performance

- Signal to Noise Ratio (SNR): Signal to Noise Ratio is usually computed in presence of a clean signal and a noise augmented distorted signal. However, in our use-case we do not have a "clean" signal to measure the generated audio against and hence we utilize WADA-SNR Kim and Stern (2008), a blind SNR estimation technique

- MOS: In order to provide a synthetic estimate for MOS we utilize DNSMOS Reddy et al. (2021) to obtain an estimate on a scale of 1-5. Section 4.1 discusses other synthetic MOS methods we explored in our study

# 3 Results

Developing on the work presented in Choudhary and Purwar (2026), we utilized the evaluation framework to measure the performance of fine tuned Qwen-0.5 billion language model backbone. Different evaluation metrics viz. Mean Opinion Score (MOS), voice similarity and latency have been rigorously investigated.

Table 3: Detailed Speaker Fine-tuning Results across HiFiTTS and LibriHeavy Datasets

| Speaker ID | Dataset | Dataset Len. used (hours) | Energy Spread | Data MOS (Mean ± Std) | Base MOS | LoRA 1000 Train Steps | △ vs Base | LoRA 5 Epochs |
|---|---|---|---|---|---|---|---|---|
| 1 | HiFiTTS | 3.523 | -29.73 ± 12.93 | 3.48 ± 0.32 | 3.832 | 3.861 | +0.029 | 3.813 |
| 2 | HiFiTTS | 3.907 | -29.34 ± 13.11 | 3.51 ± 0.37 | 3.717 | 4.141 | +0.424 | 4.106 |
| 11614 | HiFiTTS | 3.860 | -32.20 ± 13.34 | 3.30 ± 0.38 | 3.680 | 3.818 | +0.138 | 3.768 |
| 1401 | LibriHeavy | 17.590 | -32.11 ± 11.89 | 3.44 ± 0.26 | 3.659 | 3.385 | −0.274 | 3.435 |
| 1212 | LibriHeavy | 18.492 | -28.41 ± 9.91 | 3.42 ± 0.25 | 3.647 | 3.233 | −0.414 | 3.350 |
| 1259 | LibriHeavy | 13.652 | -31.18 ± 8.25 | 3.56 ± 0.29 | 3.665 | 3.664 | −0.001 | 3.614 |

In this section, we present a comprehensive analysis of the perceptual quality outcomes for Qwen-2.5 0.5B Yang et al. (2025) fine-tuned via Low-Rank Adaptation (LoRA) for per-speaker voice cloning. We focus on DNS-MOS (OVRL) Reddy et al. (2021) as the primary evaluation metric and report results for the base model, 1000 training steps and five epochs of LoRA, as well as effects of decoding-time distribution control. The evaluation is conducted over a set of six distinct speakers with varying reference audio quality. Unless specified, the audios generated for were for a 180 character long input sentence with $T=1.0$, $top\_k=50$ and the numbers reported are averaged over 5 generations. The LoRA adapters were added to the base during inference (unless specified by "Merge" in Figure 3)

## 3.1 DNS-MOS Quantitative Comparison

Table 4 summarizes the DNS-MOS (OVRL) scores obtained across different training and inference conditions. The first two columns report the reference audio quality and base model performance, respectively. Columns three and four correspond to one and five epochs of LoRA fine-tuning evaluated with temperature $T=1.0$ and top-k sampling at $k=50$. The final column reports the average DNS-MOS scores obtained with a constrained decoding regime ($T=0.8$, $k=40$) for the 1000 training steps LoRA models, averaged over five runs.

Table 4: DNS-MOS (OVRL) Comparison Across Reference, Base, and LoRA Models

| Speaker | Ref | Base | LoRA (1000 Tr. Steps) | LoRA (5 ep) | LoRA (T=0.8, k=40) |
|---|---|---|---|---|---|
| 1 | 3.481 | 3.832 | **3.861** | 3.813 | 3.739 |
| 2 | **4.145** | 3.717 | **4.141** | 4.106 | 4.048 |
| 11614 | 3.598 | 3.680 | **3.818** | 3.768 | 3.733 |
| 1401 | 3.564 | 3.659 | 3.385 | **3.435** | **3.454** |
| 1212 | **3.242** | **3.647** | 3.233 | 3.350 | **3.461** |
| 1259 | 3.772 | 3.665 | 3.664 | 3.614 | **3.754** |

Table 5: Results for LoRA vs Full Finetuning of LLM Backbone

| Speaker ID | 1 | 2 |
|---|---|---|
| MOS LoRA | 3.861 | 4.141 |
| MOS Full FT | 4.077 | 4.008 |
| Ref Audio MOS | 3.481 | 4.145 |
| Dataset Length | 3.523 | 3.907 |
| LoRA Batch Size | 4x2 | |
| Full Batch Size | 2 | |

## 3.2 Perceptual Trends and Analysis

The results in Table 4 reveal several consistent patterns. First, LoRA fine-tuning for 1000 training steps produces the most pronounced shifts in DNS-MOS relative to the base model, reflecting rapid adaptation to speaker-specific acoustic distributions. Notably, Speaker 2, with high reference quality, exhibits near-perfect alignment with its reference after 1000 training steps of LoRA, corroborating the capacity of LoRA to capture salient speaker characteristics when the reference is clean. Conversely, speakers with lower reference quality (e.g., Speaker 1212) manifest perceptual degradation after 1000 training steps, indicating that LoRA faithfully encodes both desirable and undesirable aspects of the reference distribution. With extended LoRA training (five epochs), several speakers display partial recovery in DNS-MOS, particularly those adversely affected by early adaptation. This behavior aligns with the hypothesis that the underlying pretrained LLM backbone exerts a regularizing influence, mitigating extreme deviations induced by low-quality speaker distributions. Speaker 1401, for example, experiences an increase in MOS from 1000 training steps to five epochs, suggesting stabilization as training progresses. The final column highlights the impact of decoding-time distribution shaping (reduced temperature and top-k) on perceptual quality. In this regime, low-quality speakers (e.g., Speakers 1212 and 1401) exhibit substantial MOS improvements, indicating that decoding constraints effectively suppress low-likelihood acoustic artifacts produced by early LoRA adaptation. High-quality speakers (e.g., Speaker 2) see slight MOS reductions under constrained sampling, consistent with the hypothesis that expressive nuances may occupy lower-probability regions of the learned distribution.

# 4 Discussions

## 4.1 Metrics: Mean Opinion Score (MOS)

Mean Opinion Score (MOS) is a subjective metric that is awarded by human evaluators to a sound on basis of its quality. We compared several tools for synthetic estimation of MOS viz. UTMOSv2 Baba et al. (2024), WVMOS Andreev et al. (2023), TorchAudio-Squim Kumar et al. (2023).

- **UTMOSv2:** Results calculated by UTMOSv2 across various runs for the same audio are inconsistent with significant standard deviation being observed across 10 runs for an audio file. On average, the scores for UTMOSv2 are generally lower than the other two tools (compared to WVMOS and TorchAudio-Squim)

- **WVMOS:** WVMOS is consistent across runs, however it has a bias towards ranking shorter audio clips disproportionately high with MOS score exceeding the threshold of 5 in some cases

- **TorchAudio-Squim:** Unlike UTMOSv2 and WVMOS which are blind estimation methods, TorchAudio-Squim requires a reference audio to compare a sample against and deliver MOS scores. This turns out to be unfavorable in case of TTS generated audio which do not have a reference audio to be compared against. Similar to WVMOS, Squim has a bias towards shorter audio clips and scores them disproportionately higher

DNS-MOS (OVRL) is used as the primary evaluation metric due to its consistency across runs and lack of bias towards audio length, making it suitable for evaluating TTS-generated audio without reference samples.

## 4.2 Discussion: Finetuning epochs and effect on MOS

As illustrated in Fig. 2, the dynamics of speaker-specific LoRA adaptation reveal a fundamental departure of LLM-based acoustic modeling from classical text-to-speech (TTS) paradigms. In conventional sequence-to-sequence or diffusion-based TTS systems Liu et al. (2023) Popov et al. (2021), reductions in training and validation loss are typically correlated with improvements in perceptual quality metrics such as MOS or MCD, as these models directly regress intermediate acoustic representations (e.g., mel-spectrograms or waveform samples Shen et al. (2018)) using pointwise or frame-level objectives. In contrast, the Qwen-0.5B LLM backbone employed in this work models speech generation as a conditional likelihood over learned acoustic token sequences, thereby embedding a strong pretrained acoustic prior that persists throughout speaker adaptation.

Across all speakers, the loss curves in Fig. 2 exhibit smooth and monotonic convergence, indicating stable optimization of the low-rank LoRA parameters. However, the corresponding DNS-MOS (OVRL) trends show pronounced non-monotonic behavior. For speakers with relatively clean and expressive adaptation data (e.g., Speaker 2), rapid loss reduction within the first 1000 training steps aligns with significant gains in DNS-MOS relative to the non–fine-tuned base model. Conversely, for speakers whose adaptation data contains lower perceptual quality or narrower acoustic diversity (e.g., Speaker 1212 and Speaker 1401), similarly rapid early loss reduction is accompanied by a marked drop in DNS-MOS. This indicates that early LoRA updates amplify speaker-specific acoustic modes present in the fine-tuning data, including undesirable artifacts, despite improved likelihood.

Notably, continued optimization beyond the first 1000 training steps does not result in divergence or overfitting, as evidenced by the stable loss trajectories in Fig. 2. Instead, DNS-MOS partially recovers or stabilizes for several speakers at later epochs. This behavior supports the hypothesis that the frozen Qwen-0.5B backbone reasserts its pretrained acoustic prior over time, constraining excessive deviation induced during early adaptation. Because LoRA updates are both low-rank and norm-bounded, they act as structured perturbations within the pretrained likelihood landscape, enabling refinement of speaker identity while limiting sustained amplification of low-quality acoustic modes. Such implicit regularization is notably absent in classical TTS adaptation pipelines, which often require explicit regularizers, auxiliary losses, or extensive data curation to mitigate overfitting.

Overall, the observed mismatch between likelihood-based convergence and perceptual quality highlights a defining characteristic of LLM-based TTS systems: improvements in token-level predictability do not necessarily correspond to monotonic gains in perceptual naturalness. Instead, perceptual quality emerges from the interaction between the pretrained acoustic prior, the trajectory of parameter-efficient speaker adaptation, and inference-time generation behavior.

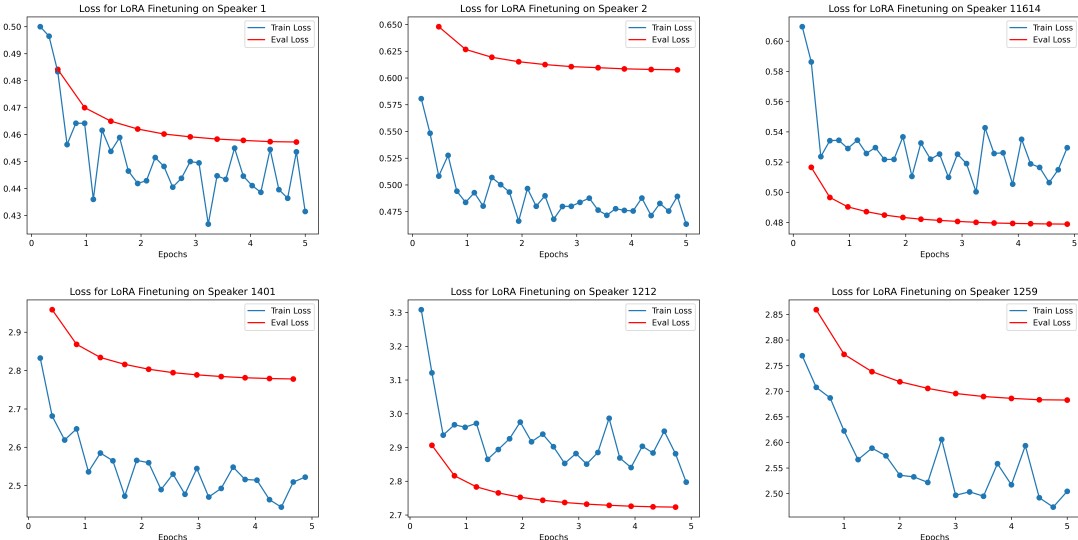

Figure 2: Plots of Training and Validation Loss at various training steps for each speaker ID. The curves indicate stochastic decrease of training loss with training steps with saturation in validation loss as we approach 4 epochs

A comparison of LoRA and full finetuning on the performance is outlined below in Table 5. The MOS scores indicated are averaged across 5 runs with same generation parameters.

### 4.3  LoRA Fine-tuning : Impact on Inference latency

We noticed a trend in Table 6 is that post finetuning the latency with longer reference audios is lesser than the latency with shorter reference audios, alongside the expected gains in MOS due to using longer reference length audio. This presents a favorable result for LoRA finetuning which facilitates use of longer cloning audios to generate audio with higher MOS.

Table 6: Latency and MOS figures for LoRA finetuning with varying reference audio length

| Dataset | HiFi TTS | | | | | |
|---|---|---|---|---|---|---|
| Speaker ID | 1 | | 2 | | 11614 | |
| Ref Audio Len. (s) | 5.84 | 8.98 | 5.02 | 7.98 | 5.21 | 9.38 |
| LoRA Inference Time (s) | 30.92 | 28.85 | 37.58 | 33.80 | 32.51 | 32.23 |
| LoRA MOS | 3.76 | 4.10 | 3.95 | 4.01 | 3.86 | 3.91 |
| Ref Audio MOS | 3.48 | 3.96 | 4.15 | 3.17 | 3.60 | 3.39 |
| Increase LoRA/Ref % | 7.95 | 3.59 | -4.63 | 26.55 | 7.34 | 15.61 |

Table 7: Latency for non-streaming generation with the neuphonic/neucodec codec for Speaker ID 1 and 2

| Speaker ID | 1 | | 2 | |
|---|---|---|---|---|
| Ref Len | 5.84 | 8.98 | 5.02 | 7.98 |
| LoRA Gen Time | 30.92 | 28.85 | 37.58 | 33.80 |
| Base Gen Time | 24.39 | 24.74 | 24.48 | 25.74 |
| LoRA Q8 Gen Time | 4.46 | 4.73 | 6.58 | 5.85 |
| Base Q8 Gen Time | 5.42 | 5.14 | 4.40 | 4.96 |

Table 8 indicates a steady increase in Speaker Similarity on fine-tuning which was an expected result considering fine-tuning was performed on a particular voice to enable better voice cloning. Similarly, increase in Signal to Noise Ratio (SNR) was also observed in cases where the cloning reference audio itself had a good SNR ($>25$) for the LoRA fine tuned model to use during inference and generated audio. In particular, for speaker ID 1259 and 1212, the reference audio used for voice cloning itself had noise, which worsened the quality of generate audio tokens leading to a decrease

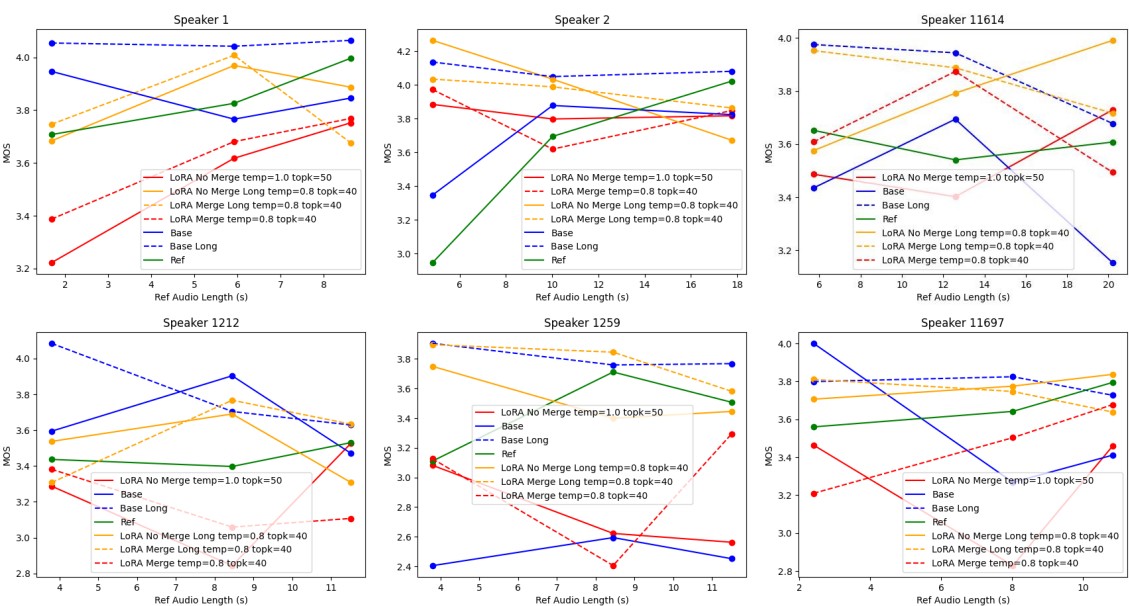

Figure 3: Plots indicating MOS Score (Y axis) of Audio Generated with varying lengths of cloning audio provided (X axis). Base refers to the standard NeuTTS model, Ref refers to the audio used for cloning. "Long" refers to generation of a longer sentence (180 characters) whereas other audios have 70 characters.

Table 8: Results of LoRA Finetuning

| Metric / Speaker ID | | 1 | 2 | 11614 | 1401 | 1212 | 1259 |
|---|---|---|---|---|---|---|---|
| Dataset | | \multicolumn HiFi TTS | | | LibriHeavy-HQ | | |
| Dataset Length (Hr) | | 3.523 | 3.907 | 3.860 | 17.590 | 18.492 | 13.652 |
| SNR | Base | 41.747 | 31.562 | 33.403 | 32.163 | 20.397 | 16.895 |
| | LoRA | 55.854 | 39.567 | 36.217 | 40.778 | 12.412 | 15.395 |
| | Ref | 31.900 | 27.047 | 24.905 | 26.791 | 22.032 | 17.630 |
| | Inc. Base/Ref % | 30.868 | 16.690 | 34.124 | 20.049 | -7.420 | -4.170 |
| | Inc. LoRA/Ref % | 75.089 | 46.288 | 45.423 | 52.207 | -43.665 | -12.679 |
| | Inc. LoRA/Base % | 33.790 | 25.364 | 8.424 | 26.787 | -39.150 | -8.879 |
| Similarity | Base | 0.732 | 0.750 | 0.727 | 0.438 | 0.511 | 0.450 |
| | LoRA | 0.818 | 0.801 | 0.820 | 0.546 | 0.610 | 0.455 |
| | Inc. LoRA/Base % | 11.799 | 6.808 | 12.871 | 24.649 | 19.472 | 1.306 |
| MOS | Base | 3.832 | 3.717 | 3.680 | 3.659 | 3.647 | 3.665 |
| | LoRA | 3.998 | 4.172 | 3.879 | 3.749 | 3.713 | 3.859 |
| | Ref | 3.481 | 4.145 | 3.598 | 3.564 | 3.242 | 3.772 |
| | Inc. Base/Ref % | 10.106 | -10.318 | 2.276 | 2.691 | 12.497 | -2.831 |
| | Inc. LoRA/Ref % | 14.861 | 0.645 | 7.818 | 5.216 | 14.537 | 2.309 |
| | Inc. LoRA/Base % | 4.318 | 12.224 | 5.418 | 2.459 | 1.813 | 5.290 |

in SNR for both base and finetuned models. Infact, the worsening of SNR was more amplified for fine tuned model, highlighting the decremental effect of fine tuning done using audio with poor SNR. For MOS values, it was observed that LoRA fine tuned model consistently delivered an increase in MOS scores over both base model and the reference audio. Contrastingly, the base model whereas lead to a decrease in MOS for speaker IDs 2 and 1259.

### 4.4 Impact of Training Data: Energy Variability, DNS-MOS Dispersion

For a better understanding of the finetuning process for various speaker IDs, we explored the metadata for the audio files. In particular, we focused on the frequency and energy statistics of the data. Frequency and energy play a key role in determining naturalness of the voice and making an audio sound human-like. Energy captures the environmental recording conditions such as background noise, microphone gain, room acoustics, distance from mic etc. Low variance in energy indicates homogeneous recording conditions (same room, same mic, same setup) and consistent speaking style (reading, not conversational) with limited acoustic diversity. Frequency on the other hand captures the characteristic tied to a particular human instead of global characteristics one would observe across human conversations.

Table 9: Impact of training-data energy statistics on DNS-MOS variability and fine-tuning outcome for speaker-specific LoRA adaptation of a Qwen-0.5B TTS backbone. While absolute energy levels (mean) show no consistent influence, higher energy variability in the training data strongly correlates with increased DNS-MOS variability and improved perceptual gains after fine-tuning.

| Speaker ID | Energy Mean | Energy Std (Training data) | DNS-MOS Std (Training data) | Fine-tuning Outcome | Observed Pattern |
|---|---|---|---|---|---|
| 2 | -29.34 | **13.11** | **0.370** | **+0.424 ×** | High energy variability + high DNS-MOS variability |
| 11614 | -32.20 | **13.34** | **0.383** | **+0.138 ×** | **Highest energy variability + highest DNS-MOS variability** |
| 1 | -29.73 | 12.93 | 0.321 | +0.029 | Medium energy and DNS-MOS variability |
| 1401 | -32.11 | 11.89 | 0.255 | **-0.274 ×** | Low energy variability + low DNS-MOS variability |
| 1212 | -28.41 | 9.91 | **0.248** | **-0.414 ×** | Second-lowest energy variability + lowest DNS-MOS variability |
| 1259 | -31.18 | **8.25** | 0.293 | -0.001 | Lowest energy variability |

Table 9 analyzes the relationship between energy statistics of speaker-specific training data, DNS-MOS variability, and perceptual outcomes obtained after LoRA fine-tuning of a Qwen-0.5B language model based TTS backbone. The results reveal a clear distinction between the roles of absolute energy level and energy variability in determining fine-tuning effectiveness.

First, the mean energy of the training data exhibits no consistent association with DNS-MOS dispersion or perceptual gains after fine-tuning. Speakers with comparable energy means (e.g., -29 dB to -32 dB) demonstrate markedly different outcomes, ranging from substantial improvement to significant degradation. This indicates that absolute loudness or global signal energy is not a reliable predictor of fine-tuning success.

In contrast, energy standard deviation emerges as a strong explanatory factor. Speakers with higher energy variability in the training data consistently exhibit larger DNS-MOS standard deviation and positive fine-tuning gains. In particular, speakers with energy standard deviation exceeding approximately 13 dB achieve the largest perceptual improvements, whereas speakers with energy standard deviation below 10 dB experience pronounced degradation. This monotonic trend suggests that energy variability serves as an effective proxy for acoustic diversity in the training corpus. The observed coupling between energy variability and DNS-MOS variability indicates that perceptual score dispersion reflects the breadth of acoustic conditions present during fine-tuning. High DNS-MOS variability implies exposure to diverse signal qualities, enabling the LoRA adaptation to better shape internal representations without overfitting to narrow acoustic manifolds. Conversely, low-variance data constrains the adaptation space, increasing susceptibility to overfitting and perceptual collapse. These findings highlight that robust fine-tuning of LLM-based TTS systems is governed by distributional diversity rather than absolute signal statistics. From a practical standpoint, enforcing minimum variability thresholds on energy-related features during data selection may be more effective than energy normalization alone.

### 4.5 Effect of context audio length

We also tried to evaluate the effect of context/ reference audio length on quality of audio output from LoRA fine tuned qwen-0.5b. For the same, we created different context length audios, by concatenating audio samples from the HiFi-TTS dataset to get longer audios, whereas for Libriheavy-HQ dataset which had longer audio samples, we sliced them to form shorter audios. Our investigation into the impact of reference audio length on voice cloning quality yielded inconsistent, speaker-dependent patterns that reveal a critical methodological insight, refer Fig. 3. Reference audio samples constructed through concatenation of shorter clips (for HiFi-TTS) or temporal slicing of longer recordings (for Libriheavy-HQ) introduced prosodic discontinuities and unnatural acoustic boundaries that confounded the relationship between context length and synthesis quality. Speakers exhibited erratic MOS score trajectories across varying reference lengths (Figure 3), with no consistent trend emerging—some speakers (e.g., Speaker 11697) showing degradation with longer references, while others (e.g., Speaker 2) demonstrated modest improvements. These mixed results suggest that the quality and naturalness of reference audio is more critical than raw duration alone. We conclude that reference audio should be sourced from single, continuous recordings to preserve prosodic coherence, rather than artificially constructed through segmentation or concatenation. Our findings indicate that high-quality (high SNR), naturally-spoken (good MOS, >3.5) reference audio from a continuous utterance is bested suited for voice cloning.

### 4.6 Finetuning on Mix Data

To evaluate the impact of training data composition (refer Appendix: Data Configuration Index) on generalisation and speaker fidelity, we compare three multi-speaker fine-tuning strategies against speaker-specific Pure FT baselines: *2+2+2 FT*, trained on 2 hours each of HiFiTTS speakers (1, 2, 11614); *1+1+1 FT*, trained on 1 hour each of the same speakers; and *Mix FT*, trained on all six speakers using 2/9th of available HiFiTTS data and 1/9th of LibriHeavy data per speaker. A critical distinction governs the interpretation of these results: the 2+2+2 and 1+1+1 models were trained exclusively on HiFiTTS speakers, making evaluation on LibriHeavy speakers (1401, 1212, 1259) a zero-shot generalisation test. On the other hand, Mix FT had exposure to all six speakers during training with less data per speaker than Pure FT.

**Zero-shot generalization of multi-speaker models**   The most striking finding is that models trained solely on HiFiTTS speakers generalise to improve MOS on completely unseen LibriHeavy speakers. Averaged across speakers 1401, 1212, and 1259, the 2+2+2 FT model achieves a DNS-MOS of 3.806 compared to 3.513 for Pure FT, a gain of +0.293 despite never encountering these speakers during training. The 1+1+1 FT model similarly improves to 3.628 (+0.114 over Pure FT). This zero-shot MOS gain arises because LibriHeavy speakers have inherently lower-quality reference audio and narrower acoustic distributions (energy std $< 12$ dB); a model overfitted to this narrow manifold amplifies its artifacts, whereas the multi-speaker model, exposed to the broader acoustic diversity of HiFiTTS, generates output of higher perceptual quality. On HiFiTTS speakers, where all models share training data, results are expectedly similar: the 2+2+2 FT model matches Pure FT closely (3.883 vs. 3.877), while 1+1+1 FT shows a small reduction to 3.824, attributable to its halved per-speaker data budget. Notably, the 2+2+2 FT model exhibits substantially lower MOS variance across all six speakers (0.008 vs. 0.052 for Pure FT), demonstrating that multi-speaker training yields significantly more consistent perceptual quality across diverse voices, even in the zero-shot setting.

**Speaker similarity and fidelity-generalization trade-off**   All multi-speaker strategies reduce speaker similarity relative to speaker-specific Pure FT, but the magnitude and interpretation differ by speaker group. For the three HiFiTTS speakers present in all training sets, the similarity degrades modestly from 0.774 (Pure FT) to 0.738 (2+2 + 2 FT), 0.735 (1+1 +1 FT) and 0.721 (Mix FT), reflecting that cost of sharing model capacity across multiple voices. For the three LibriHeavy speakers, the similarity reductions for 2+2+2 FT (0.425) and 1+1+1 FT (0.454) relative to Pure FT (0.554) are expected consequences of zero-shot prediction: these models never encountered speakers 1401, 1212, or 1259 during training, and their similarity scores reflect generalised acoustic representations rather than speaker-specific voice identity.

**Data efficiency of Mix FT**   Mix FT, using only 11-22% of the per-speaker training data of Pure FT, achieves speaker similarity of 0.721 on HiFiTTS speakers and 0.506 on LibriHeavy speakers within 7% and 9% of Pure FT respectively. Crucially, Mix FT outperforms the zero-shot 2+2+2 FT model on LibriHeavy similarity (0.506 vs. 0.425), confirming that even minimal speaker-specific data substantially improve voice fidelity over zero-shot generalization. Taken

together, these results demonstrate that distributing limited data across multiple speakers is a viable and efficient strategy for multi-speaker TTS: a single shared model trained on as little as 1/9th of per-speaker data can match within 9% of a dedicated model's similarity while serving all speakers simultaneously, without maintaining separate model weights per voice.

| Speaker ID | 1 | 2 | 11614 | 1401 | 1212 | 1259 |
|---|---|---|---|---|---|---|
| Pure FT final | 3.76 | 4.01 | 3.86 | 3.50 | 3.32 | 3.72 |
| Pure FT 1000 | 3.86 | 4.14 | 3.82 | 3.39 | 3.23 | 3.66 |
| 2+2+2 FT final | 3.96 | 3.95 | 3.738 | 3.83 | 3.74 | 3.84 |
| 2+2+2 FT 1000 | 3.95 | 4.09 | 3.77 | 3.56 | 3.69 | 3.70 |
| 2+2+2 FT 2000 | 3.79 | 4.09 | 3.75 | 3.63 | 3.71 | 3.83 |
| 1+1+1 FT final | 3.81 | 3.92 | 3.73 | 3.51 | 3.69 | 3.67 |
| 1+1+1 FT 2000 | 3.96 | 3.86 | 3.78 | 3.63 | 3.71 | 3.76 |

Table 10: Impact of different audio training data combinations used for Fine tuning Language model on MOS

| Speaker ID | 1 | 2 | 11614 | 1401 | 1212 | 1259 |
|---|---|---|---|---|---|---|
| Pure FT final | 0.790 | 0.794 | 0.737 | 0.503 | 0.678 | 0.481 |
| 2+2+2 FT final | 0.749 | 0.780 | 0.686 | 0.412 | 0.569 | 0.292 |
| 2+2+2 FT 1000 | 0.740 | 0.813 | 0.728 | 0.354 | 0.569 | 0.292 |
| 2+2+2 FT 2000 | 0.752 | 0.799 | 0.698 | 0.438 | 0.546 | 0.423 |
| 1+1+1 FT final | 0.740 | 0.786 | 0.679 | 0.396 | 0.566 | 0.399 |
| 1+1+1 FT 2000 | 0.756 | 0.785 | 0.672 | 0.422 | 0.539 | 0.401 |
| 1+1+1 FT 1000 | 0.736 | 0.790 | 0.696 | 0.421 | 0.567 | 0.443 |
| Mix FT | 0.723 | 0.768 | 0.669 | 0.442 | 0.643 | 0.432 |

Table 11: Impact of different audio training data combinations used for Fine tuning Language model on Similarity

| Speaker ID | 1 | 2 | 11614 | 1401 | 1212 | 1259 |
|---|---|---|---|---|---|---|
| Pure FT final | 57.265 | 41.640 | 51.437 | 35.504 | 19.995 | 22.704 |
| 2+2+2 FT final | 51.957 | 55.632 | 45.849 | 32.065 | 76.746 | 23.612 |
| 2+2+2 FT 1000 | 41.104 | 67.328 | 65.867 | 36.580 | 32.736 | 22.195 |
| 2+2+2 FT 2000 | 43.791 | 45.714 | 37.693 | 33.385 | 26.819 | 23.006 |
| 2+2+2 FT final | 28.871 | 50.757 | 54.411 | 32.979 | 27.112 | 21.942 |
| 1+1+1 FT 2000 | 62.6951605 | 35.7315596 | 67.6284479 | 38.3099593 | 37.0833548 | 19.5076147 |
| 1+1+1 FT 1000 | 76.6348586 | 50.3578715 | 41.9266878 | 28.4399738 | 23.8635252 | 19.8592614 |
| Mix FT | 61.6389159 | 76.2064809 | 63.3810099 | 32.7538285 | 28.4847121 | 17.0491197 |

Table 12: Impact of different audio training data combinations used for Fine tuning Language model on SNR

## 4.7 Optimizing Codec: GGUF vs non-GGUF Latency

We evaluate the impact of 8-bit GGUF quantisation on latency, throughput, and perceptual quality by comparing three configurations: the non-quantised full-precision base model (Base F32), the GGUF 8-bit quantized base model (Base Q8), and full-precision versus quantised variants of the LoRA fine-tuned model (LoRA F32 and LoRA Q8). Experiments span both non-streaming generation (Table 13) and streaming generation with chunk-wise real-time factor evaluation (Appendix: Tables 14-19), covering all six speakers and two reference audio lengths.

**Non-streaming generation: 5-7$\times$ speedup from quantisation** In non-streaming mode using the neuphonic/neucodec codec, full-precision models incur generation times that are impractical for deployment. Base F32 requires 24.4-25.7 seconds to generate audio for a single utterance regardless of reference length. On the other hand, LoRA F32 is further burdened, taking 28.9-37.6 seconds up to 54% longer than Base F32 due to the overhead of loading and

applying full-precision LoRA adapter weights at inference time. GGUF Q8 quantisation eliminates this overhead entirely: Base Q8 reduces generation time to 4.4-5.4 seconds (a 4.5-5.6× speedup over Base F32), and LoRA Q8 achieves 4.5-6.6 seconds (a 5.7-6.9× speedup over LoRA F32). Importantly, LoRA Q8 generation times are within 4% of Base Q8 for Speaker 1 across both reference lengths, demonstrating that quantised LoRA adapters add negligible computational overhead relative to the base quantised model.

A counterintuitive pattern is observed in LoRA F32 generation time (refer Table 13): it decreases as reference audio length increases, with Speaker 1 improving from 30.9 s (5.84 s ref) to 28.9 s (8.98 s ref), and Speaker 2 from 37.6 s (5.02 s ref) to 33.8 s (7.98 s ref). This suggests that longer reference audio provides richer acoustic context to the LLM backbone, enabling more confident token prediction and reducing the number of low-probability tokens that require resampling.

**Streaming generation: Q8 achieves $>2\times$ real-time; F32 does not.** In the streaming setting, quantisation has a decisive effect on sustained throughput. Full-precision LoRA F32 yields an average first-chunk latency of 0.887 s, 32.8% higher than Base Q8 (0.668 s) and an average second-chunk real-time factor (RTF) of 0.633, providing only approximately 1.5× real-time headroom. In contrast, every GGUF Q8 configuration: Base Q8, LoRA Q8, 2+2+2 FT Q8, 1+1+1 FT Q8, and Mix FT Q8, achieves a second-chunk RTF of 0.33-0.35, sustaining greater than 2× real-time throughput consistently across all six speakers. First-chunk latency for all Q8 variants falls within 0.8-5.7% of Base Q8, a range of approximately 5-38 ms in absolute terms, which is imperceptible in practice. Deploying NeuCodec Decoder on GPU, compared to CPU (with the GGUF backbone already on GPU) provides an additional 22.9% reduction in first-chunk latency (0.410 s to 0.316 s) and a 39.7% improvement in second-chunk RTF (0.521 to 0.314), pushing throughput beyond 3× real-time, with no measurable change in MOS (3.620 vs. 3.621) or speaker similarity (0.673 vs. 0.674). GPU codec placement therefore represents a zero-cost quality-neutral optimization that should be the default configuration when GPU memory permits.

**MOS and speaker similarity under quantisation** Quantisation preserves speaker similarity with high fidelity: the mean similarity change from LoRA F32 to LoRA Q8 is $+0.005$ across all six speakers (refer Appendix, Table 14-19), indicating that the dominant weight structure encoding speaker identity is robust to 8-bit precision reduction. The MOS impact is small on average ($-0.10$ DNS-MOS points from LoRA F32 to LoRA Q8) but speaker-dependent. Speakers with high acoustic energy variability in their training data: Speakers 11614 ($+0.196$) and 1259 ($+0.088$) exhibit slight MOS *gains* under quantisation, consistent with the hypothesis that quantisation noise acts as a mild regulariser against overfitting to narrow acoustic manifolds. Conversely, Speaker 1212, which has the second-lowest energy standard deviation in its training data (9.91 dB), incurs the largest quantisation MOS penalty ($-0.483$), reflecting that low-variability speakers produce LoRA weights concentrated in a narrow range that is more sensitive to 8-bit rounding error. This finding directly connects quantisation robustness to the data variability framework established in Section 4.4: the same acoustic diversity that predicts successful LoRA fine-tuning also predicts resilience to post-training quantisation. Multi-speaker Q8 configurations (2+2+2 Full Q8 and 1+1+1 Full Q8) achieve average MOS of 3.905 and 3.916 respectively both exceeding Base Q8 (3.805). This demonstrates that multi-speaker fine-tuning with GGUF quantisation can simultaneously improve perceptual quality and meet real-time latency requirements, without the per-speaker model management overhead of Pure FT.

| Sentence | S1 | | S2 | | S3 | | S4 | | S5 | |
|---|---|---|---|---|---|---|---|---|---|---|
| Env. | CPU | GPU | CPU | GPU | CPU | GPU | CPU | GPU | CPU | GPU |
| 1st Chunk Latency (s) | 0.3988 | 0.3067 | 0.4032 | 0.311 | 0.4134 | 0.3189 | 0.4195 | 0.3237 | 0.4167 | 0.3212 |
| 1st Chunk RTF | 0.7386 | 0.568 | 0.7466 | 0.576 | 0.7656 | 0.5906 | 0.7768 | 0.5994 | 0.7717 | 0.5949 |
| 2nd Chunk RTF | 0.5195 | 0.3155 | 0.5211 | 0.3124 | 0.524 | 0.3111 | 0.5217 | 0.313 | 0.5189 | 0.3194 |
| MOS | 3.19489 | 3.19882 | 3.48858 | 3.48964 | 3.78335 | 3.78371 | 3.89385 | 3.89359 | 3.73711 | 3.73772 |
| Speaker Similarity | 0.68481 | 0.68464 | 0.60134 | 0.60058 | 0.62643 | 0.62618 | 0.74191 | 0.74214 | 0.71144 | 0.71538 |
| SNR | 20.419 | 20.4192 | 65.8338 | 65.8338 | 39.0074 | 39.0007 | 55.878 | 55.8779 | 43.636 | 43.6364 |

Table 13: Impact of running codec (NeuCodec) in onnx-runtime running on CPU vs GPU for different input text sentences ( Language model in GGUF format running on GPU in all cases)

# 5 Conclusion

The quest for better voice-2-voice architectures has resulted in exploration of optimal STT and TTS techniques for cascaded/half-cascaded architectures as well as multi-modal speech-to-speech architecture. Better TTS architectures are central to cascaded/half-cascaded architectures, our work explores optimising the Language model backbone of TTS by investigating fine tuning of qwen-0.5 billion component of NeuTTS. Here are the salient findings from our investigation:

- LoRA fine-tuning of the LLM backbone yields significant MOS gains for speakers with high acoustic energy variability and DNS-MOS dispersion in their training data. Speakers with low variability show limited gains or perceptual degradation, confirming that training data diversity is the primary driver of adaptation success.

- Perceptual robustness in LLM-based TTS fine-tuning arises from exposure to heterogeneous acoustic conditions. Energy standard deviation and DNS-MOS dispersion jointly predict fine-tuning outcome, with energy std above 13 dB reliably indicating positive adaptation and below 10 dB indicating risk of perceptual collapse.

- LoRA fine-tuning consistently improves speaker similarity across all evaluated speakers, regardless of training data quality. SNR gains are observed when reference audio quality is high; conversely, noisy reference audio amplifies artefacts in the fine-tuned model, highlighting the importance of clean cloning audio.

- Training and validation loss curves are unreliable proxies for perceptual quality in LLM-based TTS. Loss improves monotonically while DNS-MOS may degrade, particularly for low-variability speakers. Checkpoint selection must therefore be guided by perceptual evaluation rather than loss convergence alone.

- Robust fine-tuning of LLM-based TTS requires distributional diversity in training audio. Acoustically homogeneous data causes the model to overfit narrow acoustic manifolds, amplifying recording artefacts. Enforcing minimum energy variability thresholds during data selection is more effective than energy normalisation alone.

- Multi-speaker LoRA fine-tuning with 1–2 hours per speaker generalises to completely unseen speakers, yielding MOS gains of +0.11 to +0.29 over speaker-specific baselines on LibriHeavy speakers never seen during training. This demonstrates that multi-speaker LLM-backbone adaptation learns transferable acoustic representations beyond the voices present in the fine-tuning set.

- Mixed-data fine-tuning across all six speakers, using only 11–22% of per-speaker data, achieves speaker similarity within 5–9% of dedicated single-speaker models. A single shared model trained on minimal per-speaker data can serve all speakers simultaneously, offering a scalable alternative to maintaining separate fine-tuned weights per voice.

- 8-bit GGUF quantisation reduces non-streaming generation time by 5.7–6.9× over full-precision LoRA and eliminates the 33–92% streaming latency overhead of F32 models. Speaker similarity is preserved with a mean change of +0.005 across all speakers, and the average MOS cost is only −0.10 DNS-MOS points. Speakers with low acoustic energy variability incur greater quantisation sensitivity, with losses up to −0.48 MOS, linking training data diversity to numerical robustness of fine-tuned weights.

- All GGUF Q8 configurations sustain second-chunk real-time factors of 0.33–0.35, exceeding 2× real-time throughput consistently across all six evaluated speakers. Full-precision LoRA F32 yields only ∼1.5× real-time headroom (RTF 0.62–0.68), which is insufficient for robust production deployment under variable system load. GGUF Q8 quantisation is therefore a practical prerequisite for real-time voice cloning, not merely an optional optimisation.

## Future Work

Fine tuning of language model backbone demonstrates improvement in performance of Text to speech model. In future, we envisage to deploy this fine tuned language model in our production voice-2-voice architecture and evaluate customer feedback.

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

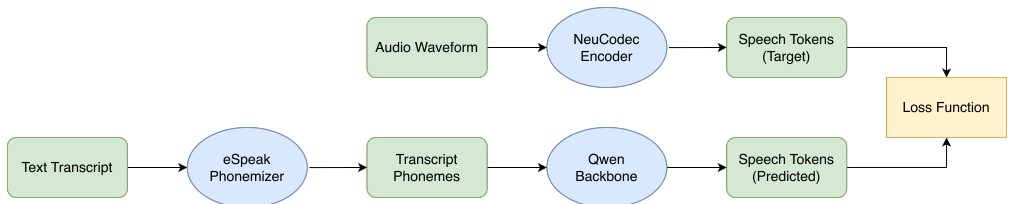

Figure 4: Pipeline for generating data for finetuning Qwen Backbone

J. Shen, R. Pang, R. J. Weiss, et al. Natural TTS synthesis by conditioning WaveNet on mel spectrogram predictions. In *Proc. IEEE Int. Conf. Acoustics, Speech and Signal Processing (ICASSP)*, pages 4779–4783, 2018. doi: 10.1109/ ICASSP.2018.8461368.

Thornbury, Bryan and Mythic Infinity Labs. Libriheavy-HQ, 2024. URL `https://huggingface.co/datasets/ mythicinfinity/libriheavy-hq`.

K. TTS. Kani TTS, https://github.com/nineninesix-ai/kani-tts, 2025. URL `https://github.com/nineninesix-ai/ kani-tts`.

C. Wang, S. Chen, et al. Neural Codec Language Models are Zero-Shot Text to Speech Synthesizers, 2023. URL `https://arxiv.org/abs/2301.02111`.

Q. . A. Yang, B. Yang, B. Zhang, et al. Qwen2.5 Technical Report, 2025. URL `https://arxiv.org/abs/2412.15115`.

Łukasz Bondaruk and J. Kubiak. LoRP-TTS: Low-Rank Personalized Text-To-Speech, 2025. URL `https://arxiv. org/abs/2502.07562`.

## Limitations

- This work has explored fine tuning on Small language models like Qwen 0.5 billion, the insights may or may not apply to Large Language models or medium language models.

- Partial utilization of per speaker data from HiFi TTS dataset. We collected first 5000 samples for each speaker. In some cases, a larger collection of data was available which was not utilized. The impact of higher longer audio data used for fine tuning on MOS and similarity in voice cloning needs further exploration with all available audio data from HiFi TTS dataset.

- We conducted a preliminary analysis of the audio data involved limited to analysis of frequency and energy distribution in place of spectrogram based analysis

- Results from fine tuning of language model backbone for TTS performed in this work may not be extendable to other tasks like text generation by language models

## Appendix

### 5.1 Data Preparation

The metadata for the dataset is maintained in a file as file-name and text transcript pair. A data preparation script reads the pairs and load the audio file into a waveform. The waveform is passed through the codec's encoder to generate the speech tokens which serve as the target for the loss function. The triplet of file-name, text transcript and the tokens is used as the input to the training pipeline.

Within the pipeline, the text transcript is converted to phonemes using eSpeak as the backend. The phonemes serve as the input to the transformer, wherein they are converted to tokens by the encoder. Augmented with the tokens for voice cloning, the transformer predicts the speech tokens for the corresponding text input tokens to generate the predicted tokens.

## Data Configuration Index

**Base Q8** NeuTTS backbone with no fine-tuning, quantised to 8-bit GGUF.

**LoRA F32** Single-speaker LoRA adapter (speaker per table caption), full float32 precision.

**LoRA Q8** Single-speaker LoRA adapter (speaker per table caption), 8-bit GGUF.

**2+2+2 1000 Q8** 2 hr each from HiFi-TTS speakers 1, 2, 11614; 1,000 training steps; 8-bit GGUF.

**2+2+2 2000 Q8** 2 hr each from HiFi-TTS speakers 1, 2, 11614; 2,000 training steps; 8-bit GGUF.

**2+2+2 Full Q8** 2 hr each from HiFi-TTS speakers 1, 2, 11614; 5 epochs (6,185 steps); 8-bit GGUF.

**1+1+1 1000 Q8** 1 hr each from HiFi-TTS speakers 1, 2, 11614; 1,000 training steps; 8-bit GGUF.

**1+1+1 Full Q8** 1 hr each from HiFi-TTS speakers 1, 2, 11614; 5 epochs (3,090 steps); 8-bit GGUF.

**Mix 1000 Q8** 2/9th HiFi-TTS data (Spk 1, 2, 11614) and 1/9th LibriHeavy-HQ data (Spk 1401, 1212, 1259); 1,000 training steps; 8-bit GGUF.

**Mix Full Q8** 2/9th HiFi-TTS data (Spk 1, 2, 11614) and 1/9th LibriHeavy-HQ data (Spk 1401, 1212, 1259); 5 epochs (2,835 steps); 8-bit GGUF.

*Note:* 2+2+2 and 1+1+1 models are trained only on HiFi-TTS speakers; evaluation on LibriHeavy-HQ speakers (1401, 1212, 1259) is zero-shot.

| - | Base Q8 | LoRAF32 | LoRAQ8 | 2+2+2 1000q8 | 2+2+2 2000q8 | 2+2+2 FullQ8 | 1+1+1 1000Q8 | 1+1+1 FullQ8 | Mix 1000Q8 | Mix FullQ8 |
|---|---|---|---|---|---|---|---|---|---|---|
| Total Time (s) | 5.33 | 10.24 | 7.57 | 5.43 | 6.26 | 5.65 | 5.94 | 5.38 | 5.97 | 6.18 |
| 1st Chunk Latency (s) | 0.52 | 0.69 | 0.50 | 0.53 | 0.53 | 0.53 | 0.50 | 0.54 | 0.54 | 0.54 |
| 1st Chunk RTF | 0.96 | 1.27 | 0.92 | 0.98 | 0.98 | 0.98 | 0.92 | 1.00 | 0.99 | 1.00 |
| 2nd Chunk RTF | 0.34 | 0.62 | 0.34 | 0.33 | 0.34 | 0.34 | 0.34 | 0.34 | 0.34 | 0.34 |
| MOS | 3.89 | 4.03 | 3.88 | 3.95 | 3.69 | 4.21 | 3.67 | 4.05 | 3.61 | 3.72 |
| Speaker Similarity | 0.76 | 0.73 | 0.74 | 0.75 | 0.78 | 0.73 | 0.79 | 0.80 | 0.59 | 0.66 |
| SNR | 27.79 | 35.00 | 42.88 | 33.31 | - | 31.01 | 40.07 | 36.87 | 29.67 | 50.44 |

Table 14: Speaker 2

| - | Base Q8 | LoRA F32 | LoRA Q8 | 2+2+2 1000q8 | 2+2+2 2000q8 | 2+2+2 FullQ8 | 1+1+1 1000Q8 | 1+1+1 FullQ8 | Mix 1000Q8 | Mix FullQ8 |
|---|---|---|---|---|---|---|---|---|---|---|
| Total Time (s) | 5.22 | 9.01 | 4.96 | 4.96 | 4.77 | 5.33 | 4.63 | 4.95 | 5.64 | 8.46 |
| 1st Chunk Latency (s) | 0.43 | 0.61 | 0.42 | 0.42 | 0.43 | 0.44 | 0.43 | 0.43 | 0.43 | 0.43 |
| 1st Chunk RTF | 0.80 | 1.13 | 0.77 | 0.78 | 0.79 | 0.81 | 0.80 | 0.79 | 0.80 | 0.79 |
| 2nd Chunk RTF | 0.33 | 0.62 | 0.33 | 0.33 | 0.33 | 0.33 | 0.34 | 0.33 | 0.33 | 0.33 |
| MOS | 4.00 | 4.01 | 3.95 | 3.88 | 3.83 | 4.04 | 3.81 | 3.94 | 3.40 | 3.69 |
| Speaker Similarity | 0.66 | 0.78 | 0.75 | 0.69 | 0.77 | 0.74 | 0.70 | 0.77 | 0.71 | 0.59 |
| SNR | 31.96 | 38.80 | 37.22 | 27.13 | 34.38 | 45.37 | 43.57 | 35.45 | 79.50 | - |

Table 15: Speaker 1

| - | Base Q8 | LoRA F32 | LoRA Q8 | 2+2+2 1000q8 | 2+2+2 2000q8 | 2+2+2 FullQ8 | 1+1+1 1000Q8 | 1+1+1 FullQ8 | Mix 1000Q8 | Mix FullQ8 |
|---|---|---|---|---|---|---|---|---|---|---|
| Total Time (s) | 5.60 | 7.47 | 5.34 | 4.98 | 4.45 | 4.24 | 4.47 | 5.22 | 7.18 | 5.47 |
| 1st Chunk Latency (s) | 0.40 | 0.60 | 0.41 | 0.40 | 0.41 | 0.41 | 0.41 | 0.41 | 0.41 | 0.40 |
| 1st Chunk RTF | 0.74 | 1.11 | 0.77 | 0.75 | 0.76 | 0.77 | 0.76 | 0.77 | 0.76 | 0.74 |
| 2nd Chunk RTF | 0.33 | 0.63 | 0.33 | 0.33 | 0.33 | 0.33 | 0.33 | 0.34 | 0.33 | 0.33 |
| MOS | 4.04 | 3.63 | 3.82 | 3.55 | 3.65 | 3.55 | 3.88 | 3.91 | 3.77 | 4.03 |
| Speaker Similarity | 0.66 | 0.74 | 0.74 | 0.68 | 0.71 | 0.77 | 0.77 | 0.70 | 0.63 | 0.68 |
| SNR | 33.55 | 45.31 | 43.26 | 81.65 | 57.00 | 61.43 | 54.96 | 51.58 | 36.26 | 59.00 |

Table 16: Speaker 11614

| - | Base Q8 | LoRA F32 | LoRA Q8 | 2+2+2 1000q8 | 2+2+2 2000q8 | 2+2+2 FullQ8 | 1+1+1 1000Q8 | 1+1+1 FullQ8 | Mix 1000Q8 | Mix FullQ8 |
|---|---|---|---|---|---|---|---|---|---|---|
| Total Time (s) | 5.42 | 9.39 | 5.33 | 5.48 | 5.24 | 4.99 | 5.36 | 5.55 | 5.97 | 5.35 |
| 1st Chunk Latency (s) | 0.96 | 1.18 | 1.06 | 1.02 | 1.06 | 1.02 | 1.00 | 0.98 | 0.99 | 0.98 |
| 1st Chunk RTF | 1.78 | 2.18 | 1.96 | 1.90 | 1.96 | 1.89 | 1.85 | 1.82 | 1.83 | 1.81 |
| 2nd Chunk RTF | 0.35 | 0.62 | 0.34 | 0.34 | 0.34 | 0.34 | 0.34 | 0.34 | 0.34 | 0.34 |
| MOS | 3.6 | 3.56 | 3.35 | 3.73 | 3.57 | 3.83 | 3.62 | 3.64 | 3.6 | 3.44 |
| Speaker Similarity | 0.43 | 0.51 | 0.50 | 0.44 | 0.48 | 0.40 | 0.43 | 0.42 | 0.50 | 0.35 |
| SNR | 33.59 | 33.36 | 41.67 | 38.36 | 38.34 | 46.75 | 38.93 | 34.91 | 10.58 | 43.16 |

Table 17: Speaker 1401

| - | Base Q8 | LoRA F32 | LoRA Q8 | 2+2+2 1000q8 | 2+2+2 2000q8 | 2+2+2 FullQ8 | 1+1+1 1000Q8 | 1+1+1 FullQ8 | Mix 1000Q8 | Mix FullQ8 |
|---|---|---|---|---|---|---|---|---|---|---|
| Total Time (s) | 7.01 | 12.85 | 6.88 | 6.33 | 8.22 | 6.34 | 7.95 | 6.03 | 5.70 | 6.99 |
| 1st Chunk Latency (s) | 0.91 | 1.20 | 0.99 | 0.94 | 0.95 | 0.95 | 0.93 | 0.94 | 0.94 | 1.00 |
| 1st Chunk RTF | 1.69 | 2.23 | 1.85 | 1.75 | 1.76 | 1.77 | 1.71 | 1.74 | 1.74 | 1.86 |
| 2nd Chunk RTF | 0.34 | 0.68 | 0.35 | 0.35 | 0.35 | 0.35 | 0.34 | 0.35 | 0.35 | 0.35 |
| MOS | 3.70 | 3.42 | 2.93 | 3.86 | 3.58 | 4.05 | 3.55 | 4.01 | 3.31 | 3.36 |
| Speaker Similarity | 0.65 | 0.64 | 0.68 | 0.57 | 0.32 | 0.50 | 0.44 | 0.51 | 0.60 | 0.60 |
| SNR | 29.69 | 30.07 | 22.10 | 26.98 | 14.65 | 39.43 | - | 33.53 | 22.86 | 24.29 |

Table 18: Speaker 1212

| - | Base Q8 | LoRA F32 | LoRA Q8 | 2+2+2 1000q8 | 2+2+2 2000q8 | 2+2+2 FullQ8 | 1+1+1 1000Q8 | 1+1+1 FullQ8 | Mix 1000Q8 | Mix FullQ8 |
|---|---|---|---|---|---|---|---|---|---|---|
| Total Time (s) | 5.49 | 8.62 | 5.68 | 5.13 | 4.91 | 5.04 | 5.14 | 5.58 | 5.10 | 5.42 |
| 1st Chunk Latency (s) | 0.79 | 1.04 | 0.85 | 0.79 | 0.80 | 0.78 | 0.78 | 0.86 | 0.86 | 0.86 |
| 1st Chunk RTF | 1.46 | 1.93 | 1.57 | 1.47 | 1.47 | 1.44 | 1.44 | 1.59 | 1.59 | 1.59 |
| 2nd Chunk RTF | 0.34 | 0.63 | 0.35 | 0.35 | 0.35 | 0.35 | 0.34 | 0.34 | 0.34 | 0.35 |
| MOS | 3.61 | 3.71 | 3.80 | 3.60 | 3.66 | 3.74 | 3.66 | 3.95 | 3.60 | 3.59 |
| Speaker Similarity | 0.41 | 0.40 | 0.42 | 0.36 | 0.41 | 0.36 | 0.37 | 0.43 | 0.43 | 0.38 |
| SNR | 17.00 | 16.62 | 19.62 | 19.92 | 17.76 | 21.43 | 19.31 | 20.28 | 17.14 | 13.00 |

Table 19: Speaker 1259

