# OpenReview forum: "When Fine-Tuning Fails and when it Generalises: Role of Data Diversity and Mixed Training in LLM-based TTS"
_TMLR — Under review for TMLR_

### Review · Reviewer_9Cqd · 2026-05-10

**Summary Of Contributions:**

This paper analyzes whether adding LoRA on an LLM-based TTS can improve the speaking quality. They focus on using LoRA to fine-tune the language model backbone and evaluate its effect on speech quality, speaker similarity, SNR, and latency across several speakers. They find that backbone LoRA can improve speech quality for some speakers, but the gains are highly dependent on the fine-tuning data. In particular, higher acoustic diversity, measured through energy variation and DNS-MOS dispersion, is associated with better adaptation, while low-diversity or noisy data can cause LoRA to learn and amplify recording artifacts. The paper also reports that training loss is not always aligned with perceptual quality, studies the effect of sampling hyperparameters, and explores mixed-speaker training and GGUF quantization for more robust and efficient inference.

**Audience:**

Yes

**Audience Explanation:**

LoRA has become a widely used and powerful adaptation method, and understanding how it behaves in LLM-based TTS systems is a meaningful question. In particular, studying whether backbone-level LoRA improves speech quality, when it fails, and how data diversity affects its performance can provide useful empirical insights for researchers working on parameter-efficient fine-tuning, speech generation, and multimodal generative models.

**Broader Impact Concerns:**

There's no concerns on the ethical implications of the work.

**Claims And Evidence:**

No

**Claims Explanation:**

* The motivation for applying LoRA to the language model backbone is not sufficiently justified. The paper does not clearly explain why this choice is better than fine-tuning other TTS components, such as speaker embeddings, acoustic decoders, or style modules.
* Some findings are relatively expected. For example, the observation that lower training or validation loss does not always imply better perceptual quality has already been observed in many other machine learning settings.
* Some results are difficult to interpret because key figures appear to be missing or improperly rendered. For example, Figure 2 and Figure 3 are referenced in the text, but the corresponding plots are not clearly shown.
* The empirical evidence is not very robust. For example, Table 9 reports results for only a small number of speakers, with each row corresponding to one specific speaker. This makes the claimed relationship between acoustic diversity and fine-tuning outcome sensitive to speaker-specific variance.

**Requested Changes:**

The authors should better justify and evaluate the main design choice of applying LoRA to the language model backbone. The current paper does not clearly explain why this is preferable to fine-tuning other TTS components, and it lacks controlled baselines against these alternatives. The empirical support and the writing for some claims should also be strengthened. For example, the acoustic diversity analysis in Table 9 is based on only a small number of speakers, so the claimed relationship between energy variability and fine-tuning outcome may be sensitive to speaker-specific variance. In addition, Figure 2 and Figure 3 appear to be missing or improperly rendered, making some results difficult to interpret. Finally, the loss-quality decoupling result should be better positioned, since lower training loss not implying better perceptual quality is a known issue in many machine learning settings. The paper should clarify what is specific to acoustic-token-based TTS.

---

> ### Author Response · Authors · 2026-05-13
> **Properly rendered images**
>
> Apologies from our end for the ill-rendered PDF. You may please refer the following arxiv link (https://arxiv.org/pdf/2603.10904) with a correctly rendered version of the paper. Figure 2, 3 are respectively on page 7, 8 of the above linked arxiv PDF
>
> We also profusely thank reviewers for their time in reviewing the manuscript. We intend to update the manuscript, but for now have shared the arxiv link of pdf with all figures.

---

### Review · Reviewer_zrmL · 2026-05-15

**Summary Of Contributions:**

This paper explores the application of LoRA fine-tuning directly to the attention layers of an LLM backbone (specifically Qwen-0.5B) within a cascaded TTS pipeline. The authors claim to identify a "loss-quality divergence" phenomenon , argue that training data energy variance acts as a strong predictor of fine-tuning success , and present multi-speaker joint training as well as GGUF quantization strategies.

Key Strengths:The core premise—fine-tuning the autoregressive language model layers rather than downstream synthesis modules—is a reasonable direction to explore.

Key Weaknesses:
1. Unacceptable Presentation and Execution Quality: The manuscript is riddled with severe formatting errors, missing figures, mismatched numbers, and incoherent text structures. It strongly resembles an unedited, raw generation from a commercial LLM.
2. Failure of Peer-Review Standard Verification: Key experimental results are broken or entirely absent. For instance, the authors heavily discuss a "Figure 4.2" and "Figure 2" to explain their core scientific claim (the loss curves) , but Figure 2 is just a broken string placeholder containing raw LaTeX-like image paths, and Figure 4.2 does not exist anywhere in the PDF.
3. Severe Data Contradictions: Quantitative metrics change inexplicably between tables for identical experimental setups.

**Audience:**

No

**Audience Explanation:**

While the theoretical topic (LLM backbone fine-tuning for speech synthesis) is of interest to the speech community, the audience cannot benefit from a paper whose empirical data is missing or structurally compromised. Publishing a manuscript with missing figures , self-contradictory tables, and zero acoustic visualization (such as spectrograms) sets a dangerous precedent for TMLR's rigorous archiving standards. The paper must be completely overhauled and re-verified before it can provide any scientific value to the community.

**Broader Impact Concerns:**

The authors are utilizing low-resource engineering frameworks (GGUF, 8-bit quantization, LoRA) to enable rapid, low-latency voice cloning of specific human speakers. This significantly lowers the barrier to entry for generating high-fidelity audio deepfakes.

The authors have completely ignored the ethical implications of this technology. It is critical that the authors include a mandatory Broader Impact Statement that discusses the dual-use nature of this work, the risks of unauthorized identity cloning, and potential technical mitigations (such as imperceptible audio watermarking or biometric defense verification).

**Claims And Evidence:**

No

**Claims Explanation:**

1. Missing Core Evidence (Figure 2 / Figure 4.2): The authors base their primary intellectual contribution—the "Loss-Quality Divergence Phenomenon"—on the visual trajectories of training and validation loss over time. However, Figure 2 is entirely broken and unrendered, presenting only a raw text string of file paths ([width=0.3]figs/losscurve1.png...). Reviewers cannot evaluate if the loss converged, if over-optimization occurred, or if the data supports the text.

2. Blatant Numeric Discrepancies: In Table 3, the LORA 5-Epoch MOS values for Speakers 1, 2, and 11614 are listed as 3.813, 4.106, and 3.768. Yet, in Table 8, under the exact same "LORA" row for the same speakers, the numbers miraculously jump to 3.998, 4.172, and 3.879. This completely undermines the credibility of the entire evaluation suite.

3. Incoherent Table Structuring: Tables 4, 5, and 8 look like corrupted data dumps. In Table 4, data rows for "Speaker ID MOS LORA" and "MOS Full FT" are randomly mashed into cells alongside speaker numbers. Table 8 uses highly unprofessional labels like "6SNR" and "3Similarity" without explaining what these multipliers mean.

**Requested Changes:**

Critical changes (Absolute prerequisites for any reconsideration):
1. Fix Missing and Broken Visual Evidence: * Figure 2 / Figure 4.2: You must properly render the training and validation loss plots. If a figure is missing or broken in review, the paper cannot be accepted.  Figure 1 & Figure 3: Ensure all image tags ([width=0.8]figs/ACL.png) are compiled into actual graphics instead of raw text placeholders.
2. Resolve Empirical Contradictions: Explain why the MOS scores for the single-speaker LoRA model drastically differ between Table 3 and Table 8. If they represent different experimental parameters, they must be labeled rigorously. If it is a copy-paste error, the integrity of all numbers must be audited.
3. Complete Table Reconstruction: Re-write Tables 4, 5, 8, and 11 using professional scientific typography (e.g., proper LaTeX booktabs). Remove arbitrary annotations like "6*SNR". Do not merge text headers directly into numerical data cells.
4. Remove AI-Generated Structural Artifacts: The text heavily suffers from lack of proofreading (e.g., source code artifacts, loose references, incomplete placeholders like "esses" at the start of line 119). The manuscript must be thoroughly rewritten to meet professional academic publishing standards.

---

> ### Author Response · Authors · 2026-05-18
> **PDF images of Fig 2,3 rendering issue**
>
> Apologies from our end for the ill-rendered PDF. You may please refer the following arxiv link (https://arxiv.org/pdf/2603.10904) with a correctly rendered version of the paper. Figure 2, 3 are respectively on page 7, 8 of the above linked arxiv PDF
>
> We also profusely thank reviewers for their time in reviewing the manuscript and highlighting the discrepancies. We intend to update the manuscript, but for now have shared the arxiv link of pdf with all figures.

---

> > ### Author Response · Authors · 2026-06-26
> > **Figures updated in  new draft of manuscript.**
> >
> > Hi,
> >
> > First we would like to thank the reviewer for their comments.
> >
> > We have addressed the issue of missing figures by uploading a new draft of manuscript.
> >
> > We will address other comments too in next few days.
> >
> > Best
> > Authors

---

### Review · Reviewer_3QYw · 2026-06-22

**Summary Of Contributions:**

The authors investigate the effect of LoRA adaptation of the LM backbone of a TTS system. They find that such an adaptation can improve text generation quality (as measured by DNSMOS) if the adaptation data is well-behaved. Specifically, the authors observed improvements when using the HiFiTTS speakers' data and degradation when using the Libriheavy data. The authors observed that shorter training is beneficial for the HiFiTTS data and is harmful for the Libriheavy data. Additionally, the authors observed that "regularization" of the inference helps voice generation on Libriheavy. The authors speculate that these results stem from the differences in perceptual quality and acoustic diversity of the recordings. Moreover, the authors analyze the effects of LoRA and quantization on latency, the effects of adaptation on mixed-speaker data, and the effects of adaptation on voice similarity.

**Strengths**
- The exact setting seems novel

**Weaknesses**
- The paper is incomplete
- The underlying motivation for the considered setup is somewhat unclear
- Some of the conclusions are too speculative
- The quality of text and tables is rough in some parts

**Audience:**

Yes

**Audience Explanation:**

I think that the paper might be interesting for the TTS community. However, the exact setup is somewhat strange.

1. The authors use data "exclusive to a single speaker in order to observe the learning of the model to a particular speaking style". However, they do not actually test this property. While the MOS results suggest that the style is consistent for any given snippet of the text, it is unclear whether the style is consistent across text snippets.
2. While the authors basically describe the differences between HiFiTTS and Libriheavy as the differences in recording quality, it might also be the case that Libriheavy texts are just harder to pronounce, or they have a different style (e.g., literature texts vs. everyday-life texts). Human studies or additional analysis are required to conclude that the main differences between the corpora are the presence of speech artifacts.
3. Generally, it might be a good idea also to test the models on a separate dataset to see how well the adapted models work in a "zero-shot" environment.

**Broader Impact Concerns:**

I do not think that the work requires a Broader Impact Statement.

**Claims And Evidence:**

No

**Claims Explanation:**

I think the paper needs some polishing and a more careful discussion.

1. The main problem with the paper currently is that all figures are missing. Since many of the results are reported in figures, it is impossible to judge the paper justly.
2. The optimizer's hyperparameters are not reported. The phrase "using Stochastic gradient descent with Adam optimizer" in Section 2.1 seems quite strange. Moreover, it is unclear how much time was devoted to hyperparameter search.
3. While the authors report using the Qwen2.5-0.5B model as an LM backbone, this model is not trained for the TTS task. Thus, the overall TTS architecture is unclear. What approach is used to produce the actual speech? How is the training loss defined? What is the actual base model?
4. The issues above make it basically impossible to assess Section 4.2. Additionally, the phrase "the frozen Qwen-0.5B backbone reasserts its pretrained acoustic prior over time" does not feel substantiated by the evidence (moreover, it does not make logical sense since the "frozen" element does not change over time).
5. Reference MOS in Table 3 and Table 4 contradict each other.
6. The conclusions at the beginning of Section 3.2 are too speculative. First, the results in Table 4 do not directly corroborate "the capacity of LoRA to capture salient speaker characteristics when the reference is clean". Given the results of Table 8, the actual similarity does increase, but rather modestly. Second, while the conclusion that "speakers with lower reference quality (e.g., Speaker 1212) manifest perceptual degradation" is acceptable, the experiment in Section 3.2 does not control for all additional factors. Specifically, using the similarity results in Table 8 again, we can conclude that the HiFiTTS data is actually closer to the base model pretraining data than the Libriheavy data. Thus, the failure of adaptation might just reveal that it is harder to adapt to out-of-distribution data. Third, I do not quite see how the results are related to "the hypothesis that the underlying pretrained LLM backbone exerts a regularizing influence".
7. Additionally, the claim that "energy standard deviation emerges as a strong explanatory factor" in Section 4.4 is too strong, given that we only have 6 points and speculate about the significance of 4 covariates: energy mean, energy std, DNSMOS mean, DNSMOS std.
8. The validity of Section 4.1 is hard to judge. The actual results are not present. Moreover, the interpretation of the results may differ. For example, it could be the case that UTMOSv2 captures the actual variability in the results. Only direct human studies can reliably tell which metric is better. Similarly, given that HiFiTTS has higher quality and is shorter, WVMOS might capture the actual difference in quality between shorter and longer recordings.
9. The text lacks polish. For instance, the references at the end of Page 1 are strange (e.g., "neu (2025)"). The references to Qwen-0.5B appear early, but the text specifies that it is the Qwen2.5-0.5B model only in Section 3. Similarly, the text uses the abbreviation MOS early, but defines it only in the first paragraph of Section 3. The first column in Table 1 duplicates the references. Table 2 lacks the dataset names. The reference to the Wespeaker embeddings is missing. The first column in Table 8 has strange artifacts. The references duplicate the links and are somewhat outdated.

**Requested Changes:**

See the claims and evidence section. At a minimum, the paper should add all missing figures, report optimizer hyperparameters, report the actual TTS architecture, and fix all typos. Additionally, the authors should be clearer about which of their findings are more and less speculative.

---

> ### Author Response · Authors · 2026-06-26
>
> Hi,
>
> First we would like to thank the reviewer for their comments.
>
> We have addressed the issue of missing figures by uploading a new draft of manuscript.
>
> We will address other comments too in next few days.
>
> Best
> Authors